# Multi-Objective Optimization of a Regional Water–Energy–Food System Considering Environmental Constraints: A Case Study of Inner Mongolia, China

**DOI:** 10.3390/ijerph17186834

**Published:** 2020-09-18

**Authors:** Junfei Chen, Tonghui Ding, Ming Li, Huimin Wang

**Affiliations:** 1Business School, Hohai University, Nanjing 211100, China; dingtonghui@hhu.edu.cn (T.D.); hmwang@hhu.edu.cn (H.W.); 2Yangtze Institute for Conservation and Development, Hohai University, Nanjing 210098, China; 3Research Institute of Jiangsu Yangtze River Conservation and High-Quality Development, Nanjing 210098, China; 4CSIRO Data61, Australian Resources Research Centre, Kensington, WA 6151, Australia; Ming.Li@data61.csiro.au; 5State Key Laboratory of Hydrology-Water Resources and Hydraulic Engineering, Hohai University, Nanjing 210098, China

**Keywords:** water–energy–food system, multi-objective programming model, environmental constraints, Inner Mongolia

## Abstract

Water, energy, and food, as the basic material resources of human production and life, play a prominent role in social and economic development. As the imbalance between the supply and demand of water, energy, and food increases, a highly sensitive and fragile relationship gradually forms among water, energy, and food. In this paper, Inner Mongolia in China is selected as a research area. Firstly, synergy theory is applied to establish the framework of a water–energy–food system. Then, a multi-objective programming model is constructed, where the objective functions are defined to minimize the integrated deviation degree and pollutant emissions of the water–energy–food system. Meanwhile, maximization of the water benefit, energy production, and food production is also considered. In addition, the model takes economy, environment, water, energy, and food as constraints. Finally, a genetic algorithm is designed for accurately assessing the most promising results. The results show that the cooperation degree of the water–energy–food system in Inner Mongolia is getting better and better, and the pollutant emission from the water–energy–food system is decreasing. In 2020, the proportion of agricultural water consumption fell by 1%, while that of industrial water consumption rose by 0.48%. The production of coal, natural gas, and power are all showing an increasing trend. Among them, the increase of natural gas production is as high as 38,947,730 tons of standard coal. However, the proportions of coal, natural gas, and power change inconsistently, where the proportions of coal and natural gas increase while that of power decreases. Corn production accounts for more than 80% of the total, which is in the eldest brother position in the food industry structure. Besides, there are differences between the planned values and optimal values of decision variables. Finally, suggestions are put forward to improve the sustainable development of water–energy–food in Inner Mongolia.

## 1. Introduction

Water resources, energy, and food, as the most important material resources to satisfy the essential needs of humans, play a crucial role in economic and social development [1]. However, with the population growth, rapid economic development, and climate change around the world, these put great pressures on the supply of water resources, energy, and food, which have gradually formed a highly sensitive and fragile relationship [2,3]. The relevant research studies have shown that the demands of global water, energy, and food are predicted to grow by 40%, 50%, and 35% by 2030, respectively, while the supply of water, energy, and food will face severe challenges owing to serious ecological environmental changes [4]. The imbalance between supply and demand of water, energy, and food has become an urgent problem to be solved. In January 2011, the Global Risk Report proposed that the “water resource–energy–food risk group” was one of the three key risk groups for the first time, emphasizing that the relationship of water resource–energy–food was very significant to the sustainable development of the regional economy and society, and only considering the optimization of a single resource would lead to unpredictable serious consequences [5]. In November of the same year, the Bonn conference firstly summarized the relationship among water, energy, and food as a nexus, and it actively explored how to balance the synergies among water, energy, and food from the perspective of the water–energy–food nexus [6].

At present, most scholars only study the nexus between two of the resources, such as the water–energy nexus [7,8,9,10,11,12,13,14], water–food nexus [15,16,17,18,19], and energy–food nexus [20,21,22,23,24,25]. According to the relevant literature, it can be concluded that for the water–energy nexus, refining, processing, and cooling of energy consume the water resources. Especially, with the rapid increase of electricity demand, the cooling water of thermal power generation also increases quickly. As for the water–energy nexus, the agriculture department is the largest water consumption department in the world, accounting for 70% of the total water consumption [26]. In terms of the energy–food nexus, energy plays an important role in the process of packaging, distribution, storage, etc., of the agricultural department.

At present, research on the water–energy–food nexus are increasing year by year, mainly focusing on the definition and challenge of the water–energy–food nexus. For example, De Amorim et al. [27] defined the water–energy–food nexus and analyzed the impact of global risks on the water–energy–food nexus. Heard et al. [28] summarized and analyzed the water–energy–food nexus in the urban system, and they thought that it was very important to analyze the water–energy–food nexus by establishing the comprehensive index system and model. Kurian [29] provided a framework to study the water–energy–food nexus and emphasized the importance of an interdisciplinary approach in the research of the water–energy–food nexus. Endo et al. [30] reviewed the water–energy–food nexus and studied the challenges and prospects of the water–energy–food nexus. Pahl-wostl [31] defined water–energy–food security from the perspective of the water–energy–food nexus. Chi et al. [32] believed that although progress had been made, there were still limitations in the research of the water–energy–food nexus that faced four challenges in the future, including the definition of a system boundary, the uncertainty related to modeling, the analysis limitation of the internal mechanism of the nexus, and the evaluation of system performance. Zhi et al. [33] built a “water–energy–food” symbiotic system framework based on the symbiosis theory and put forward the regional system adaptation concept of “water–energy–food” from the perspectives of stability, coordination, and sustainability. In addition, some scholars adopted correlative models to conduct quantitative researches on the water–energy–food nexus. For instance, Bazilian et al. [34] established the Climate- Land-Energy-Water (CLEW) model to study the water–energy–food nexus. Ziv et al. [35] adopted the method of fuzzy cognitive mapping (FCM) to analyze the water–energy–food nexus and found that the energy-related elements had the greatest impact on the water–energy–food nexus. Chen et al. [36] constructed an evaluation index system of the vulnerability and coordination of the water–energy–food system based on the Pressure-State-Response (PSR) model and used a cloud matter-element model to evaluate the degree of coordination, taking the northwest region as a case study. Meanwhile, some scholars have made use of case studies to research the water–energy–food nexus. For example, Taniguchi et al. [37] studied the water–energy–food nexus by taking 20 cases in the Asia-Pacific region as the research subject. Owen et al. [38] took the UK as an example and analyzed the interactions among water, energy, and food by using the input–output method. Additional, some scholars carried out the studies on Inner Mongolia from the water–energy–food nexus perspective. For example, Chen et al. [39] used the Slacks-Based Measure (SBM) super-efficiency model and Malmquist-Luenberger (ML) index analysis method to evaluate the total factor productivity (TFP) of the water–energy–food system in Inner Mongolia; they applied the Tobit model to study the influential factors of the water–energy–food system and found that there was a serious difference in TFP between Inner Mongolia cites. Furthermore, the mechanization level and degree of opening up have positive effects on the TFP, while the enterprise scale and the output of the third industry have negative effects on the TFP. Shang et al. [40] quantified the temporal patterns of socioeconomic growth, energy consumption, and food and water footprints of Inner Mongolia from 1987 to 2015, and they found that water resource use increased four-fold, energy consumption increased approximately seven-fold, and large areas of natural grasslands were converted to agricultural, industrial, and urban land use, which were exacerbated by large-scale coal production.

Now, some research studies on the optimization of water–energy–food have also been paid more and more attention by scholars. For example, Hang [41] proposed a systematic mathematical modeling-based approach for designing local production systems and developed a superstructure-based optimization model specifically for design of the food–energy–water nexus in a local context, which considered each supply subsystem individually and allows insights into the potential interactions between them. Zhang and Vesselinov [42] put forward a comprehensive evaluation method to optimize the production of water, energy, and food in the study of water–energy–food security, with the goal of minimizing energy supply, water supply, food production and the comprehensive cost of carbon dioxide emissions. Zhang et al. [43] used an integrated water–food–energy nexus model and optimization method to combine real-time drought monitoring with irrigation management to overcome the negative impact of agricultural drought. Karan et al. [44] built the water–energy–food system and proposed a stochastic mathematical model to forecast demand and output, which was applied to both dry and humid environments. Mo et al. [45] combined multi-objective programming, nonlinear programming, and intuitive fuzzy number, and constructed a comprehensive Agricultural Water-Energy-Food Sustainable Management (AWEFSM) optimization model of water–energy–food considering the constraints of limited water resources and energy in the agricultural system, taking northwest China as an example for empirical study. Mo et al. [46] established an optimization model for sustainable management of the water–energy–food nexus under uncertainty conditions to achieve maximum economic benefits and the minimum environmental impact. Therefore, a number of optimization models, such as the linear programming model, nonlinear programming model, dynamic programming model, and stochastic programming model, have been used to promote optimization research to the maximization or minimization of certain objectives [47]. Since the optimization of research on the water–energy–food system depends on various aspects, such as economy, environment, water, energy, food, and so on, the multi-objective programming model, which is able to solve multiple conflicting objectives functions, can be used to solve the optimization problem.

In conclusion, there are few scholars who pay close attention to the optimization studies of the water–energy–food system from a regional perspective. In addition, the field of studies about the water–energy–food system is still in its infancy in China. Therefore, in this paper, Inner Mongolia in China was selected as a research area to demonstrate a multi-objective optimization study of a water–energy–food system. Firstly, synergy theory was applied to establish the framework of the water–energy–food system. Then, the multi-objective programming model was constructed with objectives and constraints. Finally, a genetic algorithm was designed for accurately assessing the most promising results. The research can provide a theoretical framework and technical support for the comprehensive management and sustainable development of a water–energy–food system in Inner Mongolia in the future. The paper is organized as follows. Section 2 introduces the study area and data sources. Section 3 describes the methodology. The main results and discussion are presented in Section 4. Section 5 gives the conclusions of the study.

## 2. Study Area and Data Sources

Inner Mongolia (37°24′–53°23′ N, 97°12′–126°04′ E) is located in the north of China (Figure 1), including nine prefectural cities and three alliances, namely, Hohhot, Baotou, Hulunbuir, Wuhai, Chifeng, Tongliao, Ulaan Chal, Baynnur, Ordos, Xing’an, Xilin Gol, and Alax. Inner Mongolia is a vast territory with an area of 1,183,000 km^2^. The terrain of Inner Mongolia stretches from the northeast to the southwest in a narrow and slender shape, with a linear distance of 2400 km from east to west and a span of 1700 km from north to south.

The spatial and temporal distribution of water resources in Inner Mongolia is very uneven, which does not adapt to the distribution of population and farmland. Inner Mongolia is a major province of energy production in China. It not only has abundant energy resources such as coal, oil, and natural gas, but also has an optimistic prospect for the development and utilization of new energy sources such as wind energy, solar energy, and bio-energy. There are 808 billion tons of confirmed coal reserves in Inner Mongolia, ranking the first in China. The reserve of petroleum geological resources is 614 million tons, and that of natural gas is 1.67 trillion m^3^, ranking the third in China. Hence, Inner Mongolia has provided an important pillar for China’s energy development. Food grown in Inner Mongolia mainly includes corn, wheat, potato, soybean, millet, and sorghum. The corn production in Inner Mongolia accounts for nearly 80% of all food production, which has gradually developed into an “eldest brother” in the food structure in Inner Mongolia. Therefore, Inner Mongolia is a typical region to study the optimization research of a water–energy–food system, and it will promote the sustainable development of a water–energy–food system in Inner Mongolia.

This study constructed an optimization model of a water–energy–food system in Inner Mongolia based on a multi-objective programming model with 2017 as the level year and 2020 as the planning year. The data mainly included a variety of parameters in this paper. The sources of datasets included the Statistical Yearbook of Inner Mongolia, Economic and Social Development Bulletins of Inner Mongolia, Environmental Bulletin of Inner Mongolia, 13th Five-Year Plan, and Water Resources Bulletin of Inner Mongolia.

## 3. Methodology 

### 3.1. Synergy Theory

The physicist Haakon firstly put forward and systematically expounded the synergy theory. He pointed out that the system was composed of many subsystems and thought that the interaction between each subsystem produced synergetic effects to achieve the ordered state under a certain condition. Synergy theory referred to the overall synergistic effect generated by the interaction of various subsystems in a composite system [48,49]. Based on the concept of synergy theory, the regional water–energy–food complex system, which can be divided into a water subsystem, energy subsystem, and food subsystem, was constructed in this paper [50,51,52,53]. In the water–energy–food complex system, there was a synergistic effect between subsystems. Besides, a synergistic effect means that when each subsystem or each element of the water–energy–food system acts in coordination, the effectiveness of the whole system is far greater than the sum of the effectiveness when each of these subsystems or elements acts alone to achieve the effect of 1 + 1 > 2. 

From the perspective of resource flows, the regional water resources subsystem, energy subsystem, and food subsystem are closely linked. Firstly, in Inner Mongolia, the water resources subsystem is composed of the underground water, surface water, rain water, and reuse water. As for the energy subsystem, it is mainly constituted by coal, oil, wind energy, biomass energy, electric energy, and so on. In terms of the food subsystem, it mainly includes wheat, corn, rice, beans, etc. Then, water resources are primarily utilized in agriculture, industry, and life departments after purification and transport. In addition, the extraction, purification, transport, utilization, discharge, and reuse of water resources need the support of electric energy. Meanwhile, water resources are provided to the process of mining, transporting, and using coal, oil, natural gas, electric energy, and so on. The cultivation and utilization of food needs to consume a large amount of water resources. Besides, the transport of food also needs a supply of electric energy. Some kinds of food can be converted into biomass energy to become a form of energy. Therefore, it can be concluded that the water resources subsystem, energy subsystem, and food subsystem interact to form a water–energy–food complex system, the goal of which is to realize the sustainable development of regional water, energy, and food and maximize economic, social, and environmental benefit [54,55].

From the perspective of synergistic effects, the interactions within the regional water–energy–food system are strong. Efficient water resource allocation and water conservancy project arrangement can provide stable and reliable water supply for energy and food production. Scientific energy scale, structure, and distribution can help improve water resources allocation capacity and improve food production efficiency. A reasonable scale of agricultural irrigation land and food planting structure can reduce water and energy consumption and provide diversified food options. Through the internal optimization and external coordination of the water–energy–food system, the whole composite system comes into a virtuous cycle and achieves the overall coordination [56]. Therefore, the synergistic effects of the regional water–energy–food system based on synergy theory refer to a virtuous cycle of sustainable development through the interactions among subsystems in a complex large system. Each subsystem or each component in the subsystem cooperates with and supports each other to achieve the overall goal. Under the synergistic effect, the optimization of the regional water–energy–food system is to realize the overall optimization through the internal optimization of the water resource subsystem, energy subsystem, and food subsystem as well as the mutual feedback linkage between each other under the overall goal of ensuring the harmonious development of the regional economy and society. The framework of a regional water–energy–food system based on synergy theory is shown in Figure 2.

### 3.2. Establish Multi-Objective Programming Model

#### 3.2.1. The Principle of Multi-Objective Programming Model

The multi-objective programming model was firstly proposed by Charles and Cooper in 1961, which was a linear programming model composed of three elements that included a decision variable, objective function, and constraint [57]. The multi-objective programming model can be formulated by Equations (1) and (2).
(1)$$\max y=\left\{f_{1}(x),f_{2}(x),\cdots ,f_{k}(x)\right\}$$
(2)$$\begin{array}{l} s.t.\;\;\;g_{j}(x)\leq 0,j=1,2,\cdots ,p\\ h_{k}(x)=0,k=1,2,\cdots ,q\; \end{array}$$
where x denotes the decision variable. $f_{1}(x),\;f_{2}(x),\;\cdots ,\;f_{k}(x)$ represents the objective functions, respectively. $g_{j}(x)$ and $h_{k}(x)$ are the constraints. 

#### 3.2.2. Decision Variables

In this paper, the decision variables were firstly determined when constructing the multi-objective programming model of a regional water–energy–food system. Suppose $X=(x_{1},x_{2},\cdots ,x_{n})$ represents the decision variables for the optimization model of a regional water–energy–food system. The meaning and unit of each decision variable are shown in Table 1.

#### 3.2.3. Objective Function

In this paper, a regional water–energy–food system collaborative optimization model with a total score structure was established. The overall goal was to achieve the overall coordination of the water–energy–food system—that is, to control the movement direction of each subsystem of water resources, energy, and food, so as to minimize the comprehensive deviation of planned goals of the regional water–energy–food system. The sub-goal is to construct the optimization goal of water resources, energy, and food subsystems, namely to realize the maximization of economic benefit of water resources, energy, and food production, respectively. Therefore, objective functions, which represent the overall goal and different subsystem objectives respectively, were included in the model.

##### Integrated Deviation Degree Objective Function

The overall goal is to minimize the integrated deviation degree of a regional water–energy–food system by coordinating water resources distribution, energy development, and food production in various departments. The integrated deviation degree of a water–energy–food system includes the deviation degree of the water resource subsystem, deviation degree of the energy subsystem, and deviation degree of the food subsystem. The deviation degree of the water resources subsystem refers to the deviation between the planned and actual values of water resources used for all kinds of energy and food production. The deviation degree of the energy subsystem refers to the deviation between planned and actual energy production. The deviation degree of the food subsystem refers to the deviation between the planned and actual food production. The smaller the deviation degree, the higher the cooperation degree of a regional water–energy–food system. Thus, the equation can be converted to:(3)$$\min f_{1}(x)=\sum_{i=1}^{9}\left[\omega_{i}(\frac{S_{i}^{*}-x_{i}}{S_{i}^{*}})\right]\;$$
where $f_{1}(x)$ denotes the comprehensive coordination degree of the water–energy–food system. $S_{i}^{*}\;(i=1,2,3)$ are the planned agricultural water consumption, planned industrial water consumption, and planned domestic water consumption (m^3^), respectively. $S_{i}^{*}(i=4,5,6)$ represent the planned coal production, planned natural gas production, and planned power production (10^4^ tons of standard coal). $S_{i}^{*}(i=7,8,9)$ denote the planned wheat production, planned corn production, and planned soybean production (10^4^ ton). $\omega_{i}(i=1,2,\cdots ,9)$ are the extent to which water, energy, and food contribute to the achievement of the overall goal, which is determined by the analytical hierarchy process (AHP) in accordance with the guidelines of efficient utilization of water resources, ensuring energy and food security.

##### Water Benefit Objective Function

The objective of water resources distribution is to achieve the maximum economic benefit through a rational allocation of water resources and optimization of water supply structure. The objective function is calculated by Equation (4),
(4)$$\max f_{2}(x)=\sum_{i=1}^{3}x_{i}(BC_{i}-FC_{i})$$
where $BC_{1}$ is the efficiency coefficient of agricultural water (yuan/m^3^ and yuan is the monetary unit in China). $FC_{1}$ represents the cost coefficient of agricultural water (yuan/m^3^). $BC_{2}$ denotes the efficiency coefficient of industrial water (yuan/m^3^). $FC_{2}$ is the cost coefficient of industrial water (yuan/m^3^). $BC_{3}$ is the efficiency coefficient of domestic water (yuan/m^3^). $FC_{3}$ represents the cost coefficient of domestic water (yuan/m^3^).

##### Energy Production Objective Function

The objective of energy optimization is to maximize energy production by setting up the energy structure rationally and arranging the energy scale scientifically. The objective function is given as Equation (5),
(5)$$\max f_{3}(x)=\sum_{i=4}^{6}x_{i}.$$

##### Food Production Objective Function

The aim of food optimization is to maximize regional food production through rational planning and structural selection of food. The objective function is shown as Equation (6),
(6)$$\max f_{4}(x)=\sum_{i=7}^{9}x_{i}\;.$$

##### Pollutant Emissions Objective Function

The goal is to minimize the pollutant emissions from a regional energy subsystem by optimizing the regional energy production structure. The objective function is obtained by Equation (7),
(7)$$\min f_{5}(x)=\sum_{i=4}^{6}x_{i}(\alpha_{i}+\beta_{i})$$
where $\alpha_{4}$ indicates the sulfur dioxide emission coefficient of coal production per unit. $\beta_{4}$ represents the smoke (powder) emission coefficient of coal production per unit. $\alpha_{5}$ represents the sulfur dioxide emission coefficient of natural gas production per unit. $\beta_{5}$ is the smoke (powder) emission coefficient of natural gas production per unit. $\alpha_{6}$ represents the sulfur dioxide emission coefficient of power production per unit. $\beta_{6}$ denotes the smoke (powder) emission coefficient of power production per unit (10^4^ ton/10^4^ tons of standard coal).

#### 3.2.4. Constraints

##### Economic Constraint

Economic constraint states that the costs of energy and food production must be less than the maximum costs. These constraints were established to ensure that costs were minimized.
(8)$$C^{e}\sum_{i=4}^{6}x_{i}\leq C_{\max}^{e}$$
(9)$$C^{f}\sum_{i=7}^{9}x_{i}\leq C_{\max}^{f}$$
where $C^{e}$ and $C_{\max}^{e}$ are the costs of energy production per unit and the maximum energy production, respectively (10^4^ yuan/10^4^ tons of standard coal and 10^4^ yuan). $C^{f}$ and $C_{\max}^{f}$ represent the costs of food production per unit and the maximum food production, respectively (10^4^ yuan/10^4^ ton and 10^4^ yuan).

##### Environmental Constraint

The environmental constraint should not exceed the pollutant emissions stipulated by the sustainable development standard for the sum of sulfur dioxide and smoke (powder) emissions from the water–energy–food system.
(10)$$\sum_{i=4}^{6}x_{i}(\alpha_{i}+\beta_{i})\leq GDP*PDP$$
where PDP is the pollutant emissions per unit of GDP (10^4^ ton/10^4^ yuan).

##### Water Production Constraint

Water production constraint stipulates that the water consumption of the agricultural, industrial, and domestic departments shall not be greater than the water provision of the agricultural, industrial, and domestic departments.
(11)$$\sum_{i=1}^{3}x_{i}\leq WU_{\max}$$
(12)$$AWU_{\min}\leq x_{1}\leq AWU_{\max}$$
(13)$$IWU_{\min}\leq x_{2}\leq IWU_{\max}$$
(14)$$DWU_{\min}\leq x_{3}\leq DWU_{\max}$$
where $WU_{\max}$ denotes the maximum exploitable and utilized water resources (m^3^). $AWU_{\min}$ is the minimum agriculture water consumption (m^3^). $AWU_{\max}$ expresses the maximum agriculture water consumption (m^3^). $IWU_{\min}$ is the minimum industrial water consumption (m^3^). $IWU_{\max}$ is the maximum industrial water consumption (m^3^). $DWU_{\min}$ is the minimum domestic water consumption (m^3^). $DWU_{\max}$ is the maximum domestic water consumption (m^3^).

##### Energy Production Constraint

Energy production constraint stipulates that the production of the coal, natural gas, and power should not be less than the minimum energy production requirements. Meanwhile, regional energy production should meet the energy self-sufficiency rate.
(15)$$x_{4}\geq CP_{\min}$$
(16)$$x_{5}\geq GP_{\min}$$
(17)$$x_{6}\geq TP_{\min}$$
(18)$$\frac{\sum_{i=4}^{6}x_{i}}{TEC}\geq ESR$$
where $CP_{\min}$ represents the minimum coal production (10^4^ tons of standard coal). $GP_{\min}$ represents the minimum natural gas production (10^4^ tons of standard coal). $TP_{\min}$ is the minimum power production (tons of standard coal). TEC represents the total energy consumption (10^4^ tons of standard coal). ESR is the energy self-sufficiency rate (%).

##### Food Production Constraint 

The regional food production should not be lower than the regional minimum food production requirement, and the regional food production per capita shall not be inferior to the minimum food production per capita. At the same time, food production should meet the food self-sufficiency rate.
(19)$$\sum_{i=7}^{9}x_{i}\geq GY_{\min}$$
(20)$$\frac{\sum_{i=7}^{9}x_{i}}{TP}\geq PFO_{\min}$$
(21)$$\frac{\sum_{i=7}^{9}x_{i}}{UAY}\geq CA$$
(22)$$\frac{\sum_{x=7}^{9}x_{i}}{TFC}\geq FSR$$

##### Non-Negative Constraint 

(23)$$x_{i}\geq 0(i=1,2,\cdots ,9)$$
where $GY_{\min}$ is the minimum food production (10^4^ ton). $PFO_{\min}$ is the minimum food production per capita (10^4^ ton per person). UAY is food production per unit of cultivated area (10^4^ ton/ha). CA is the guaranteed area of food farmland (ha). TFC is the total food consumption (10^4^ ton). FSR is the food self-sufficiency rate (%).

### 3.3. Genetic Algorithm

A genetic algorithm was put forward by professor Holland. Genetic algorithms are based on techniques inspired by evolutionary biology such as inheritance, selection, crossover, and mutation [58]. The approximate optimal solution to the problem is obtained through continuous iteration, which is suitable for dealing with complex and nonlinear problems [59].

An algorithm begins with a set of chromosomes, which is called a population. The initial population of chromosomes is randomly generated according to a given population size. The iterative process of the genetic algorithm is that, firstly, the objective function is evaluated by calculating the fitness value of each population. Then, a set of chromosomes are selected to mate with. Finally, the solutions in the mating pool perform genetic manipulation: crossover and mutation. Crossover is a process in which a new generation shares many positive characteristics with parents, while mutation is a process in which a group of solutions is randomly selected and changed in its original state. The generational process is repeated until a termination condition has been reached [50]. Therefore, it is significant to apply a genetic algorithm to calculate the results of the multi-objective programming model of a regional water–energy–food system. 

In this study, the weight coefficient transformation method is adopted to solve the genetic algorithm. Assume that the sub-objective function of a multi-objective programming model is given a weight, where the weight represents the importance of the objective function. Then, the linear weighted sum of each sub-objective function is as follows:(24)$$u=\sum_{i=1}^{n}\omega_{i}\cdot f(x_{i})$$
where u indicates the evaluation function of the multi-objective programming model; then, the multi-objective programming problem will be transformed into the single-objective programming problem, which can be solved by a single-objective genetic algorithm.

## 4. Results and Discussion

### 4.1. Parameter Estimation

The parameters for the multi-objective programming model can be divided into eight parts in this study. They are socioeconomic parameters, cost parameters, environment parameters, water resources parameters, water efficiency parameters, water cost parameters, energy parameters, and food parameters, respectively.

#### 4.1.1. Socioeconomic Parameters

Socioeconomic parameters included GDP and total population. These parameter values were obtained mainly from the Statistical Yearbook of Inner Mongolia and the 13th Five-year Plan for Inner Mongolia. Table 2 provided detailed information.

#### 4.1.2. Cost Parameters

The cost parameters mainly were composed of energy costs and food costs. As for the costs of energy production, they were constituted by production costs and environmental costs. Therefore, the costs of coal production per unit, natural gas production per unit, and power production per unit are shown in Table 2. According to the cost coefficient of energy production in Inner Mongolia, the maximum energy production cost was forecasted in 2020. Meanwhile, related food production costs are shown in Table 2.

#### 4.1.3. Environmental Parameters

Referring to the management practice of resource conservation and comprehensive utilization, the pollutant emissions coefficients of energy could be obtained. Specific information was shown in Table 3.

#### 4.1.4. Water Resources Parameters

The total water consumption could be obtained by the 13th Five-Year Plan in Inner Mongolia. Then, according to the Statistical Yearbook of Inner Mongolia, it can be predicted that the largest water consumption of the agricultural, industrial, and domestic departments accounted for 75%, 9%, and 6% of the total water consumption, respectively. Meanwhile, the minimum water consumption of the agricultural, industrial, and domestic departments accounted for 55%, 8% and 5%, separately. In addition, the planned water consumption of the agricultural, industrial and domestic departments was obtained through the 13th Five-Year Plan. According to the Inner Mongolia water resources bulletin, the maximum exploitable and utilized water resources can be estimated in Inner Mongolia in 2020. The specific figures are shown in the Table 4. 

#### 4.1.5. Water Efficiency Coefficient Parameters

The water use departments include the agricultural department, industrial department, domestic department, etc., and there are differences in the calculation methods of the water efficiency coefficient for different water use departments.

For the efficiency coefficient of agricultural water, it can be defined by Equation (25),
(25)$$B_{1}=\frac{\beta_{1}\times BAW}{IQ}.$$

Here, $B_{1}$ represents the efficiency coefficient of agricultural water (10^4^ yuan/m^3^). $\beta_{1}$ is the allocation coefficient of agricultural water conservancy ranging from 0.2 to 0.6. BAW represents the irrigation efficiency of agricultural water (10^4^ yuan/mu). IQ is the irrigation quota (m^3^/mu).

According to the Statistical Yearbook of Inner Mongolia, relevant data could be obtained from 2013 to 2017 in Inner Mongolia. Then, the irrigation efficiency of agricultural water was calculated by the ratio of agricultural production value (yuan) to irrigation area (mu) (Table 5). The irrigation efficiency and irrigation quota in 2020 were estimated by using the time-series trend prediction method. Therefore, the efficiency coefficient of agricultural water in 2020 can be calculated.

In terms of the efficiency coefficient of industrial water, it can be obtained by Equation (26),
(26)$$B_{2}=\beta_{2}\times \frac{\left(\frac{Q_{I}}{W}\right)}{Q_{I}}.$$

Here, $B_{2}$ represents the efficiency coefficient of industrial water (yuan/m^3^). $\beta_{2}$ denotes the efficiency sharing coefficient of industrial water supply taking 0.11. $Q_{I}$ is industrial water consumption. W is the water consumption of industrial added value (m^3^/yuan).

According to the 13th Five-Year Plan and the Water Resources Bulletin in Inner Mongolia, it can be expected that the water consumption of industrial added value will be reduced by 20% by 2020. Thus, the efficiency coefficient of industrial water in Inner Mongolia can be gained in 2020 (Table 6).

Since the domestic water consumption did not directly generate benefits, the efficiency coefficient of domestic water was 0 yuan/m^3^ (Table 6).

#### 4.1.6. Water Cost Coefficient Parameters

The cost coefficients are different in different water departments, so the specific cost coefficient can be obtained by referring to the water fee collection standard of Inner Mongolia (as shown in Table 6).

#### 4.1.7. Energy Parameters

According to the 13th Five-Year Plan for the energy industry in Inner Mongolia and the Energy Strategic Action Plan (2014–2020), we can get the relevant energy parameters, which are demonstrated in Table 7.

#### 4.1.8. Food Parameters

According to the 13th Five-Year Plan of Inner Mongolia and the National Medium-Long Term Plan for Food Security (2008–2020), the relevant food parameters can be obtained, which are shown in Table 8.

### 4.2. Results of Multi-Objective Programming Model

The optimization model of a regional water–energy–food system was designed with the objectives of minimizing the integrated deviation degree and the pollutant emissions from the energy subsystem (10^4^ tons of standard coal), maximizing the water economic benefit (yuan), the energy production (10^4^ tons of standard coal), and food production (10^4^ ton). Then, the model took economy, environment, water resources, energy, and food as constraints. Therefore, through the overall control and coordination of water resources, energy, and food, the optimization model promoted the efficient utilization of water resources, energy, and food, and it realized the integrated optimization layout of the regional water–energy–food system. The objective values and decision variable values of the optimization model were accurately assessed by genetic algorithm (as shown in Table 9 and Table 10).

As seen in Table 9, it can be revealed that the integrated deviation degree of the water–energy–food system is 0.2001 in Inner Mongolia in 2020, indicating that with the improvement of management and technical level, and the deepening understanding of the importance of the water–energy–food nexus, the cooperation degree of a regional water–energy–food system is getting better and better, and the efficiency of resource allocation and utilization has been gradually improved in the processes of water–energy, water–food, and energy–food transformation. The economic benefit of water resources increased by 26,548,966,000 yuan from 2017 to 2020, indicating that the overall efficiency of water resources is promoted by the rational allocation of water resources in the agricultural, industrial, and domestic departments. The energy production increased by 70.05% compared with the starting year, indicating that there is still a large space for the development of energy in Inner Mongolia. Food production has also improved substantially over the previous year. The pollutant emission from the water–energy–food system has decreased compared with 2017, indicating that with the in-depth development of the concept of green development, Inner Mongolia is paying more and more attention to ecological and environmental protection and is achieving remarkable results. Table 10 shows the consumption of water, energy, and food in 2020. As for water resources, the water consumptions of agricultural, industrial, and domestic departments are 15,824,000.000 m^3^, 1,899,000.000 m^3^ and 1,266,000.000 m^3^ in 2020, respectively. Then, the production of coal, natural gas, and power is 821,445,000, 39,000,000 and 72,784,635 tons of standard coal in 2020, respectively. Meanwhile, the production of corn, wheat, and soybean in Inner Mongolia is 31,605,233, 2,100,058, and 533,723 ton in 2020, respectively.

According to the results of the multi-objective programming model, the comparison diagrams of water consumption, energy production, and food production in 2017 and 2020 are drawn in this paper (as shown in Figure 3). According to the optimal values of decision variables calculated by the optimization model (as shown in Table 10) and planned values based on the 13th Five-Year Plan of Inner Mongolia, it can be found there are differences between the planned values and optimal values of decision variables (as shown in Table 11).

## 5. Discussion

### 5.1. Discussion of Results

As seen from Figure 3, from the subsystem perspective, as for the water subsystem, the specific water consumption of different departments can be obtained from Table 10 in 2020. It can be summarized that the agricultural department is the biggest water consumer, which accounts for nearly 83% of the total, but the proportion falls by 1%, followed by the industrial department, and finally the domestic department. It can be seen that the consumption of water resources for food production has decreased, but not by much. Meanwhile, the water consumption of the industrial department and domestic department shows increasing trends. This is because as the main food production province, the agricultural development in Inner Mongolia aggravated the contradiction between supply and demand of water resources. On the one hand, the planting area and irrigation area developed too fast. On the other hand, the planting area of water-consuming crops such as corn increased too fast. Therefore, the agricultural water consumption in Inner Mongolia is relatively high. However, due to effective measures regarding the sustainable utilization of water resources, such as finding alternative water sources and developing water-saving irrigation technologies, the agricultural water consumption in Inner Mongolia has decreased somewhat. Additionally, Inner Mongolia is rich in mineral resources, the development of which has brought a rapid increase in water consumption. Meanwhile, the industrial water-saving technologies and facilities in Inner Mongolia are backward, and some enterprises use extensive water, which resulted in an increase of 0.48% in industrial water consumption in Inner Mongolia. Finally, in recent years, Inner Mongolia has made great progress in water resources management and ecological construction, but the management foundation is still weak. Furthermore, the red lines of groundwater exploitation and utilization have not yet been refined and decomposed to all cities, resulting in disordered and extensive groundwater exploitation in some areas.

It can be concluded from Figure 3 that the total energy production increased by 70.05% compared with the original year. Among the energy sources, coal production increased by 66.19% in 2020, the proportion of which dropped from 90.49% to 88.02%. Natural gas production increased the most, by as much as 38,947,730 tons of standard coal in 2020, the proportion of which rose from 0.00046% to 4.18%. Meanwhile, power production increased by 33.51%, and the proportion decreased from 9.98% to 7.8%. Therefore, the production of coal, natural gas, and power in Inner Mongolia all show increasing trends, but the proportions change inconsistently. The changes of coal, natural gas, and power production are the results of various energy documents issued by governments at all levels in Inner Mongolia in recent years to promote the transformation of Inner Mongolia’s energy structure to clean and low carbon. At present, coal is still the main energy, and the coal-based energy structure will remain for a long time in Inner Mongolia. However, the productive process of coal not only needs abundant water resources, but it also would produce a large amount of environmental pollutants, which is destructive to the environment, so Inner Mongolia should appropriately control the development of coal. Natural gas, which belongs to clean energy, consumes less water and produces fewer environmental pollutants in the process of production, and the proportion of production and consumption of clean energy is increasing year by year, with huge development potential in Inner Mongolia. In recent years, Inner Mongolia’s power supply capacity continues to be strong, and the total amount of power production increases year by year. The power structure of Inner Mongolia is still mainly thermal power generation, which is supplemented by wind power, hydropower, and solar power generation. However, thermal power production consumes an abundant amount of water and has a great impact on the environment, while hydroelectric power expends almost no water. Meanwhile, wind power is renewable energy and does not generate pollution. Therefore, its power structure is constantly optimized, while the proportion of thermal power is on the decline, wind power and solar power generation development momentum is good and has huge potential. Additionally, it can be revealed that the corn production accounted for more than 80% of the total, which holds the eldest brother position in the food industry structure. Among them, wheat production increased by a large margin, while soybean production decreased by 67.18%. Corn is a water-consuming crop in Inner Mongolia, and the corn production in Inner Mongolia was as high as 80% in 2020, indicating the huge amount of water resources used to produce corn. As a result, the agricultural department became the largest water consumer sector. 

### 5.2. Discussion of the Difference of the Planned Value and Optimal Value

It can be revealed from Table 11 that the optimal values of agricultural, industrial, and domestic water consumptions are all higher than the planned values. This is because Inner Mongolia has implemented the Implementation Measures for the Pilot Reform of Water Resource Tax in Inner Mongolia Autonomous Region since December 1, 2017. Through the change of water resource fee tax, a water resource tax system with reasonable regulation and efficient management was established to effectively play the role of tax lever, reasonably adjust water demand and water use behavior, enhance awareness of water-saving, and improve water efficiency. Next, in order to encourage the development of agricultural industries such as planting and breeding, some enterprises and institutions engaged in agricultural production were exempted from water resource tax within the prescribed quota, which led to the increase of agricultural water consumption. Therefore, the reason why the optimal values of water resources in Inner Mongolia in 2020 are higher than the planned values is due to a series of water-related measures taken by the government in recent years. As for the energy subsystem, with the exception of power production, the optimal production of coal and natural gas are all lower than planned values. This is due to the oversupply of coal production in Inner Mongolia in the past, resulting in excess coal capacity. Therefore, in recent years, Inner Mongolia has optimized the distribution of energy production and stabilized the coal production capacity by strictly implementing the control requirements of “three regions and three lines”, and it has formulated the coal development distribution plan of the whole region. No new coal mines will be opened in the core area of the grassland, thus reducing the optimal value of coal. In addition, Inner Mongolia actively develops clean energy and gives preferential policies and financial support to the clean energy industry, greatly increasing the production of natural gas and other clean energy. However, due to the restriction of clean energy development technology, the production of natural gas has increased but still not reached the planned value. As for the food subsystem, the optimal corn production is higher than the planned value, and the optimal soybean production is inferior to the planned value, while the optimal wheat production differs little from the planned value. This is because in recent years, the corn market continues to go up, and the price is high, while that of soybean is low, which is leading to the phenomenon where corn is “the single largest food” in the food planting structure of Inner Mongolia. Meanwhile, the planting area of soybean keeps shrinking, and the production gradually decreases.

## 6. Conclusions

In this paper, synergy theory was applied to establish the framework of a water–energy–food system, and the multi-objective programming model was adopted to construct the optimization model of a water–energy–food system in Inner Mongolia, which was designed with the goals of minimizing the integrated deviation degree and pollutant emissions from the energy subsystem, maximizing the water economic benefit, energy production, and food production, taking the economy, environment, water, energy, and food as constraints. Then, a genetic algorithm was designed for accurately assessing the most promising results. Therefore, this paper draws the main conclusions and proposes policy suggestions to improve the sustainable development of the water–energy–food system in Inner Mongolia.

Firstly, it can be seen that the cooperation degree of a regional water–energy–food system is getting better and better, because the pollutant emission from the water–energy–food system is reducing. Besides, the agricultural department remains the largest water consumer, although its proportion declines, while the water consumption of the industrial department and domestic department shows an increasing trend. There is still a large space to save water in agricultural, industrial, and domestic departments. Therefore, the government should continue to vigorously promote water conservation in agriculture, industry, and life to comprehensively improve the efficiency and benefits of water resources utilization. First of all, Inner Mongolia should comprehensively promote the development of water-saving agriculture through a combination of engineering, administrative, economic, technological, and managerial measures, increase financial support for water-saving agricultural practices, and ensure the implementation of water-saving agricultural irrigation. Next, Inner Mongolia is supposed to vigorously promote water-saving transformation in industry, encourage enterprises to increase water-saving in production technology, advocate recycling, and improve water utilization efficiency. Third, Inner Mongolia should promote the renovation of water-saving facilities in urban areas and do a good job in the construction and renovation of sewage treatment and reuse facilities.

Secondly, it can be revealed that the production of coal, natural gas, and power in Inner Mongolia are all showing an increasing trend. However, the proportions of coal, natural gas, and power change inconsistently, where the proportions of coal and natural gas increase, while that of power decreases. In addition, there are substantial increases of food production in Inner Mongolia, but the food planting structure is not reasonable. Among them, corn production is in the eldest brother position, while the proportion of wheat and soybean production is low. Thus, Inner Mongolia firstly should give full play to the basic supporting role of coal, encourage the use of new technologies, new equipment, and new processes, and reduce the pollutant emission based on its resource advantages dominated by coal. Then, the construction scale of clean energy such as wind energy, solar energy, and biomass energy should be rationally planned and drawn up. Furthermore, Inner Mongolia is supposed to promote the diversification of its food planting structure. It is valuable to boost the planting of other crops such as beans, cereals, and potato to meet social demand.

Lastly, it can be concluded that there are differences between the planned values and optimal values of decision variables. The optimal values of water consumption of all departments are higher than the planned values, the same as the optimal corn production. Besides, the optimal production of coal and natural gas is inferior to planned values, the same as the optimal soybean production. Therefore, in the future, Inner Mongolia should allocate a water quota for various industries in a scientific way, strictly control water consumption for industry, reduce agricultural irrigation water, and moderately guarantee water for life and ecology. Then, Inner Mongolia should adjust the agricultural planting structure according to its own water resource endowment conditions. On the one hand, the planting area of water-consuming food crops such as corn should be appropriately reduced. On the other hand, that of drought-tolerant crops such as soybean should be increased. Finally, Inner Mongolia should adhere to the concept of green development and vigorously develop natural gas, wind energy, and other clean energy, taking building a green, low-carbon, safe, and efficient energy system as the development goal. 

All in all, the present study identified the framework of the water–energy–food system and carried out optimization research from the overall perspective of the water–energy–food system. Therefore, this paper is of great value to the research of the water–energy–food system. From the perspective of methodology, a multi-objective programming model has been successfully applied to reflect the complexities of the water–energy–food system and it has been proven to be effective in the case study, so this methodology has reference value in the future research of water–energy–food optimization. From the perspective of synergy theory, at present, very few research studies construct the framework of a water–energy–food system based on the synergetic theory, and few of them take into account the synergetic effects within a water–energy–food system. Hence, this study provides a new research idea for water–energy–food academic research in the future. However, due to the lack of data, water pollutant emission and solid waste discharge as important factors were not considered in the objective functions. Meanwhile, although eight parameters have been taken into account in our model, there are also some other factors that should be considered to make the results more complete. Hence, the study has its own limitations, which calls for improvement in future related studies. First, a more reasonable model that considers the more comprehensive objective functions and constraints for a water–energy–food system is desired. Second, an application of optimization research of a water–energy–food system in other typical regions would be recommended in future research.

## Figures and Tables

**Figure 1 ijerph-17-06834-f001:**
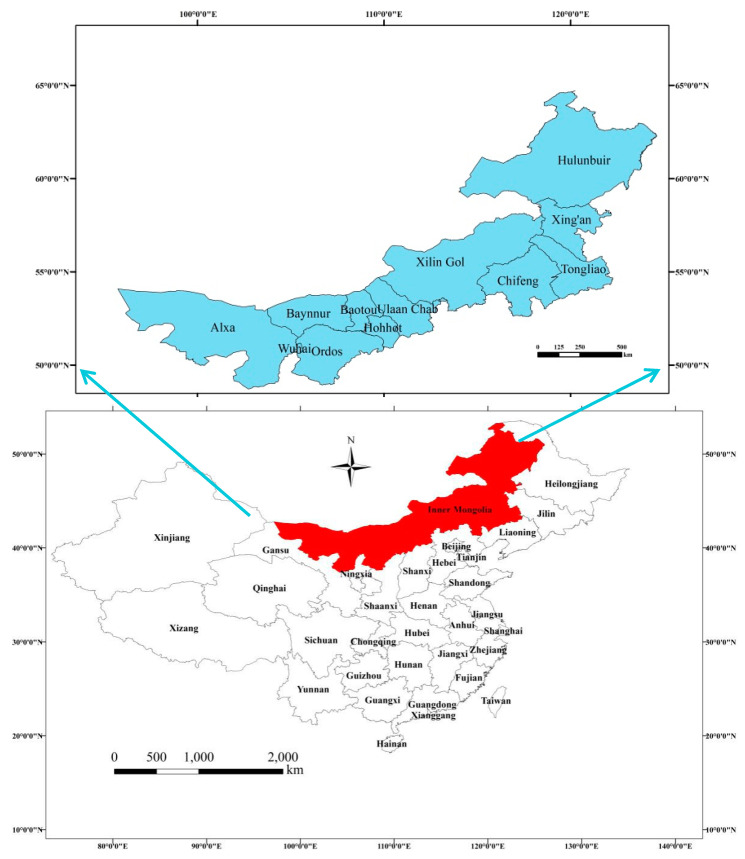
Geographical position and districts of Inner Mongolia, China.

**Figure 2 ijerph-17-06834-f002:**
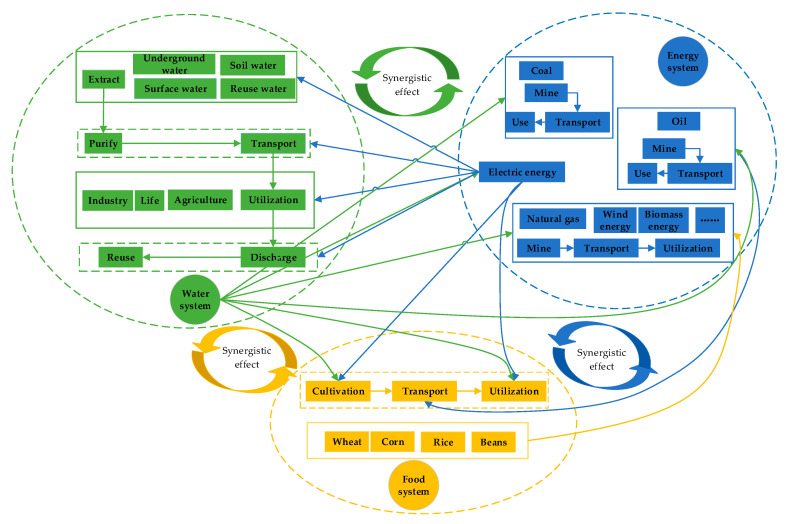
The framework of a regional water–energy–food system based on synergy theory in Inner Mongolia.

**Figure 3 ijerph-17-06834-f003:**
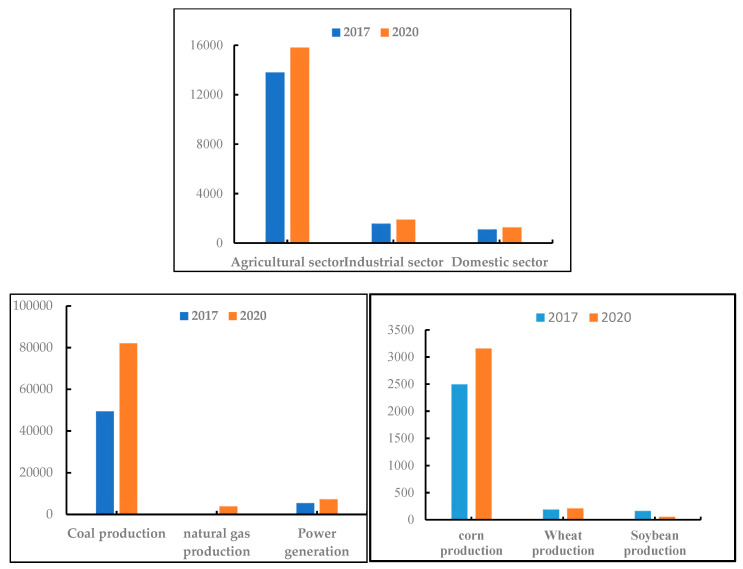
Comparison diagram of water consumption of different departments as well as different energy and food production in Inner Mongolia.

**Table 1 ijerph-17-06834-t001:** The meaning and unit of each decision variable of the multi-objective programming model of a regional water–energy–food system.

Decision Variable	Meaning	Unit
$x_{1}$	Agricultural water consumption	10^4^ ton
$x_{2}$	Industrial water consumption	10^4^ ton
$x_{3}$	Domestic water consumption	10^4^ ton
$x_{4}$	Coal production	10^4^ tons of standard coal
$x_{5}$	Natural gas production	10^4^ tons of standard coal
$x_{6}$	Power production	10^4^ tons of standard coal
$x_{7}$	Corn production	10^4^ ton
$x_{8}$	Wheat production	10^4^ ton
$x_{9}$	Soybean production	10^4^ ton

**Table 2 ijerph-17-06834-t002:** Socioeconomic and cost parameters of a water–energy–food system in Inner Mongolia.

1st Parameter	2nd Parameter	Value	Unit
Socioeconomic parameters	GDP	233,440.000	10^4^ yuan
	Total population	2660	10^4^ population
Cost parameters	Costs of coal production per unit	807.9	10^4^ yuan/10^4^ tons of standard coal
	Costs of natural gas production per unit	862.7	10^4^ yuan/10^4^ tons of standard coal
	Costs of power production per unit	2440	10^4^ yuan/10^4^ tons of standard coal
	Maximum costs of energy production	99,204,832.42	10^4^ yuan
	Costs of corn production per unit	1057	10^4^ yuan/10^4^ ton
	Costs of wheat production per unit	1787.6	10^4^ yuan/10^4^ ton
	Costs of soybean production per unit	3796	10^4^ yuan/10^4^ ton
	Maximum costs of food production	3,918,700	10^4^ yuan

**Table 3 ijerph-17-06834-t003:** Environment parameters of a water–energy–food system in Inner Mongolia.

1st Parameter		2nd Parameter	Value	Unit
**Environmental parameters**	Emission coefficient of coal production per unit	Sulfur dioxide emission coefficient	0.02	10^4^ ton/10^4^ tons of standard coal
		Smoke (powder) dust emission coefficient	0.0131	10^4^ ton/10^4^ tons of standard coal
	Emission coefficient of natural gas production per unit	Sulfur dioxide emission coefficient	$4.7\times 10^{-4}$	10^4^ ton/10^4^ tons of standard coal
		Smoke (powder) dust emission coefficient	$2.1\times 10^{-4}$	10^4^ ton/10^4^ tons of standard coal
	Emission coefficient of power production per unit	Pollutant emission coefficient per unit power	0	10^4^ ton/10^4^ tons of standard coal
	Pollutant emissions per unit GDP	Pollutant emission per unit GDP	$3.1246\times 10^{-4}$	10^4^ ton/10^4^ yuan

**Table 4 ijerph-17-06834-t004:** Water resources parameters of a water–energy–food system in Inner Mongolia.

1st Parameter	2nd Parameter	Value	Unit
Water resources parameters	Planned agricultural water consumption	$1.3715\times 10^{10}$	m^3^
	Planned industrial water consumption	$1.7935\times 10^{9}$	m^3^
	Planned domestic water consumption	$1.1605\times 10^{9}$	m^3^
	Maximum exploitable and utilized water resources	$4.5184\times 10^{10}$	m^3^
	Minimum agricultural water consumption	$1.1605\times 10^{10}$	m^3^
	Maximum agricultural water consumption	$1.5825\times 10^{10}$	m^3^
	Minimum industrial water consumption	$1.688\times 10^{9}$	m^3^
	Maximum industrial water consumption	$1.899\times 10^{9}$	m^3^
	Minimum domestic water consumption	$1.055\times 10^{9}$	m^3^
	Maximum domestic water consumption	$1.266\times 10^{9}$	m^3^

**Table 5 ijerph-17-06834-t005:** Agricultural production data in Inner Mongolia from 2013 to 2017.

Year	Production Value	Irrigated Area	Irrigation Efficiency	Irrigation Quota
2013	1368.88	4436.64	3.241	264
2014	1457.94	4517.82	3.099	322
2015	1474.54	4630.35	3.140	327
2016	1477.56	4697.295	3.179	305
2017	1434.73	4762.245	3.319	308

**Table 6 ijerph-17-06834-t006:** Water efficiency coefficient and cost coefficient of the water–energy–food system in Inner Mongolia.

1st Parameter	2nd Parameter	Value	Unit
Water efficiency coefficient	Efficiency coefficient of agricultural water	4.489	yuan/m^3^
	Efficiency coefficient of Industrial water	58.51	yuan/m^3^
	Efficiency coefficient of domestic water	0	yuan/m^3^
Water cost coefficient	Cost coefficient of agricultural water	0.11	yuan/m^3^
	Cost coefficient of industrial water	4.57	yuan/m^3^
	Cost coefficient of domestic water	0.1	yuan/m^3^

**Table 7 ijerph-17-06834-t007:** Energy parameters of the water–energy–food system in Inner Mongolia.

1st Parameter	2nd Parameter	Value	Unit
Energy parameters	Planned coal production	92,859	10^4^ tons of standard coal
	Planned natural gas production	7448	10^4^ tons of standard coal
	Planned power production	7278.1380	10^4^ tons of standard coal
	Minimum coal production	82,144.5	10^4^ tons of standard coal
	Minimum natural gas production	3900	10^4^ tons of standard coal
	Minimum generating capacity	6175.3888	10^4^ tons of standard coal
	Total energy consumption	22,500	10^4^ tons of standard coal
	Energy self-sufficiency rate	85	%

**Table 8 ijerph-17-06834-t008:** Food parameters of a water–energy–food system in Inner Mongolia.

1st Parameter	2nd Parameter	Value	Unit
Food parameters	Planned corn production	2670	10^4^ ton
	Planned wheat production	210	10^4^ ton
	Planned soybean production	190	10^4^ ton
	Minimum food production	2750	10^4^ ton
	Food production per unit cultivated area	$5.148\times 10^{-4}$	10^4^ ton/ha
	minimum food production per capita	$0.4\times 10^{-4}$	10^4^ ton
	Cultivated area for food	$5.336\times 10^{6}$	ha
	Total food consumption	374.528	10^4^ ton
	Food self-sufficiency rate	95	%

**Table 9 ijerph-17-06834-t009:** Objective values of the optimization model of the water–energy–food system in 2020.

Year	Integrated Deviation Degree	Water Economic Benefit	Energy Production	Food Production	Pollutant Emissions
2020	0.2001	171,598,756,000	93,322.9635	3423.9014	2721.635

**Table 10 ijerph-17-06834-t010:** Decision variable values of the optimization model of the water–energy–food system in 2020.

Decision Variable	Value	Unit
Agricultural water consumption	15,824,000.000	m^3^
Industrial water consumption	1,899,000.000	m^3^
Domestic water consumption	1,266,000.000	m^3^
Coal production	82,144.5	10^4^ tons of standard coal
Natural gas production	3900	10^4^ tons of standard coal
Power production	7278.4635	10^4^ tons of standard coal
Corn production	3160.5233	10^4^ ton
Wheat production	210.0058	10^4^ ton
Soybean production	53.3723	10^4^ ton

**Table 11 ijerph-17-06834-t011:** The planned value and optimal value of decision variable in 2020.

Decision Variable	Planned Value	Optimal Value
Agricultural water consumption	13,715,000.000	15,824,000.000
Industrial water consumption	1,793,500.000	1,899,000.000
Domestic water consumption	1,160,500.000	1,266,000.000
Coal production	92,859	82,144.5
natural gas production	7448	3900
Power production	7278.1380	7278.4635
Corn production	2670	3160.5233
Wheat production	210	210.0058
Soybean production	190	53.3723

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
