# Peer review of "Multi-Objective Optimization of a Regional Water–Energy–Food System Considering Environmental Constraints: A Case Study of Inner Mongolia, China"

_ijerph, 2020, doi:10.3390/ijerph17186834_

Round 1

Reviewer 1 Report

This paper analyses food energy water nexus in the Inner Mongolia as a case study. The major tool is mathematical optimization. However several clarifications need to be made before acceptance regarding the deatils of the methods applied. Please consider the following comments for improving paper quality

  1. In the abstract is not clear when you mention the amount of natural gas, and you give the number in tons of coal.
  2. Introduction. There are currently many studies focusing on optimization and quantification of the three components FEW nexus. One of the first studies using optimization considering food, energy and water simultaneously is: Designing integrated local production systems: a study on the food-energy-water system, MYL Hang, E Martinez-Hernandez, M Leach and A Yang. J of Cleaner Production 135, 1065-1084, 2016. Among others you could review please.
  3. It is not clear what concepts or methods from 'synergy theory' where applied. It seems this is not really used and the description is too general. If this is not a widely accepted and established methodology please avoid it as the description just says what is widely known about the nexus.
  4. The paper needs to clearly state or explain what are the specific synergies considered in the system modelling, or what synergies were found important from your results. This is more valuable than wether you use a theory to define your system components. 
  5. The whole section 3.1 only describes in general terms each method, but it does not provide the specificsof how these methods where applied to your problem. You dont need to put definitions or histrory of optimization models or genetic algorithms, thede can be found in a text book. 
  6. Revise the use of some English words, as it read like you are using yield as synonim of production, these are not the same in the paper context. Some sentences are not well connected. 
  7.  The optimization problem is stated twice, please just leave one set of equations. Also, please describe each of the equations variables right after each equation, rather than at the end of a group of equations. This improves readibility and following the model approach.
  8. Explain the difference between the decision variablrs and the planned quantities used. In adittion please explain how the planned quantities were established, one could think that these quantities need to be the result of your model. 
  9. Plese clearly define the concept of integrated deviation and why this is thr major objetive function.
  10. How subjectivitiy of the weigthing used in the integrated deviation would affect model results? Do you need sensitivy analyses?
  11. Please detail the approach followed to solve the multiobjective problem. 
  12. The parameter estimation is not a result, and it seems they were not estimated but taken from statistics.
  13. The results are describ3d but not discussed. Please rather than repeating the values in the tables in the text, explain and discuss your results. For example what a deviation of 0.2 means or implies.
  14. Mentioning the numbers obtained is not a conclusion. 
  15. The whole discussion and conclusions need to be rewritten.

Reviewer 2 Report

This paper applied combined methods such as synergy theory to establish the WEF system, the multi-objective programming model to construct the optimization model of the WEF system, then a generic algorithm for assessing the result. Furthermore, this paper proposes policy suggestions to improve sustainable development of the WEF system in Inner Mongolia. The results show the changes in water consumption, energy production, and food yield between 2017 and 2020.

  1. According to the aims and scope of the Journal of Environmental Research and Public Health, it targets mainly the interdisciplinary area of environmental health sciences and public health. This means papers submitted to the journal might focus more on the interrelationships between environmental health and the quality of life. Judging from this point of view, the content of the article fundamentally doesn’t meet the journal aims or its scope. Therefore, the authors could also discuss interrelationships between environment and human health.
  2. In the Introduction section, the authors reviewed the research of WEF nexus from different angles. I would suggest that the authors review more the previous studies on inner Mongolia from the WEF nexus perspective.
  3. The authors constructed the optimization model of WEF system based on a multi-objective programming model with 2017 as the level year and forecasted 2020 (L145~). Please clarify the reason why you selected those years, since the year 2020 is currently ongoing.
  4. Please explain more about Figure 2 from the point of the synergy system as the figure doesn’t visualize inclusion of the synergy theory, but only explains the WEF system. Furthermore, I was wondering if the figure explains the general WEF system at the regional scale or specifically that in Inner Mongolia.
  5. There are not clear differences between Results and Discussion, and Conclusion sections. You could discuss the limitations or future challenges of your studies in Discussion or Conclusion sections. Moreover, the authors made several policy recommendations both in Results and Discussion, and Conclusion sections. However, the authors do not mention any concrete current policy, legislations, and intuitions that the authors target. Please explain more about governance and management aspect if you would like to change or improve them relating to WFF resources, systems, and sectors in Inner Mongolia based on the results.
  6. “coil” should be changed to “coal” (L402).
  7. The authors should avoid the use of ‘ a lot’ (412) in an academic paper and could provide a number with the reference.
  8. Although the authors applied combined methods to establish the optimization model of WEF system, the authors summarize the water, energy, and food separately (except irrigation) in the Conclusion section. You could explain and discuss your results focusing more on interrelationships or interdependencies, and synergies relationships among WEF (because you use the synergy theory), if you intend to use the frameworks of the WEF nexus and/or the WEF nexus approach.

Round 2

Reviewer 1 Report

The authors have made an improvement following reviewers' comments. The only remaining comment is that the response to the comment on what principles, concepts or methods from synergy theory are applied is not satisfactory. Also, the references provided (48 and 49) cannot be accessed and it would be difficult to review such a theory if it the book is in Chinese, therefore I suggest either removing this part, or providing a more accessible reference.

Author Response

Dear reviewer: Please see the attachment.

Reviewer 2 Report

Point 4.

The authors added the words “synergistic effect” in Figure 2, and further provide an explanation of synergy system in the content (L 199-207) based on synergy theory to establish the water-energy-food system in this paper. In addition, the authors answered that the Figure 2 explains general WEF system at the regional scale.

However, the Figure 2 looks too general and simple to explain general WEF system including synergy system at the regional scale, though the WEF nexus system is more comprehensive and complicated even in specific case study areas. Please visit the URL below and see the papers that address the WEF nexus systems.

https://www.mdpi.com/2073-4441/10/9/1245

https://www.mdpi.com/2073-4441/8/10/425

I suggest the authors focus the WEF system in your study site (Inner Mongolia) to explain the WEF system including the synergy theory more in details in Figure 2. Furthermore, the explanation you added (L 199-207) seems to be explanation more on optimization of resources.

The defining synergy effect in the framework of WEF nexus is one of the hot topics to discuss in the nexus academic community, therefore, it is important to define and/or explain the synergy effect in your paper. Thus, you could review the previous studies on synergy effects in the framework of WEF nexus, and revise the part from the point of synergy.

Point 5.

I recommend the authors create Results and Discussion sections separately to distinguish the results from your discussion, as it is difficult for readers to ascertain the exact results of the Multi-Objective Programming Model. After showing the statistics and numbers in Results section, you could discuss the possible causes and factors of the results, and recommendations in Discussion section. In addition, please provide statistics and/or numbers about synergetic effects in Results section.

Point 5.

The authors concluded that “application of optimization research of water-energy-food system in other typical 608 regions would be recommended in the future research” in the end of Occlusion section. Please explain more about how the study contributes to the WEF nexus study in the WEF nexus academic community, for example, from the point of synergy system or methodology.

In my understanding, this paper is valuable for academic readers as well as other stakeholders in the WEF nexus community, as there are few articles that discuss the WEF nexus in Inner Mongolia. Moreover, the method combining the synergy theory, the multi-objective programming model, and a generic algorithm is one of the features of your WEF nexus study.

Author Response

Dear reviewer:

Round 3

Reviewer 2 Report

  1. Figure 2.

Please add “in Inner Mongolia” in the title of the Figure 2, and explain the direction of arrows.

  1. L172

Please add the reference for Haakon.

  1. L176-182

Please add the source of the theory or concept in the list of references.

  1. L389

The tile of the section could be “4. Results”. Then you could create a new section “5. Discussion” on L468. However, Figure 3, 4 and Table 11 seem to be results of your study.

  1. L677-685

Please add the source of the information in the list of references.

The potential articles for nexus methods are as follows.

https://iopscience.iop.org/article/10.1088/1748-9326/aaa9c6/meta

https://www.sciencedirect.com/science/article/pii/S246858441930056X

Please add the other references for the synergetic theory.

Author Response

Dear reviewer,
